# SEMI-SUPERVISED LEARNING BY COACHING

## ABSTRACT

Recent semi-supervised learning (SSL) methods often have a teacher to train a student in order to propagate labels from labeled data to unlabeled data. We argue that a weakness of these methods is that the teacher does not learn from the student's mistakes during the course of student's learning. To address this weakness, we introduce Coaching, a framework where a teacher generates pseudo labels for unlabeled data, from which a student will learn and the student's performance on labeled data will be used as reward to train the teacher using policy gradient.

Our experiments show that Coaching significantly improves over state-of-the-art SSL baselines. For instance, on CIFAR-10, with only 4,000 labeled examples, a WideResNet-28-2 trained by Coaching achieves $96.11\%$ accuracy, which is better than $94.9\%$ achieved by the same architecture trained with 45,000 labeled. On ImageNet with 10% labeled examples, Coaching trains a ResNet-50 to $72.94\%$ top-1 accuracy, comfortably outperforming the existing state-of-the-art by more than $4\%$. Coaching also scales successfully to the high data regime with full ImageNet. Specifically, with additional 9 million unlabeled images from OpenImages, Coaching trains a ResNet-50 to $82.34\%$ top-1 accuracy, setting a new state-of-the-art for the architecture on ImageNet without using extra labeled data.[1]

## 1 INTRODUCTION

Professional players in competitive sports such as chess, tennis, or swimming often have coaches to help improving their performance. Although coaches typically do not play as well as the players, they observe the players and provide instructions to improve the players' performance. Modern semi-supervised learning (SSL) algorithms do not follow this strategy. They instead have a teacher model that generates pseudo labels for unlabeled data, from which a student model learns by imitation (*e.g.,* Lee (2013); Tarvainen & Valpola (2017); Laine & Aila (2017)). A weakness of these methods is that the teacher does not adjust itself based on the student's performance and cannot adapt to make the student better over time, unlike professional sport coaches develop their players.

Here, we propose a new semi-supervised learning method, called *Coaching* as shown in Figure 1, where the teacher learns throughout the course of student's training. In our method, a teacher generates pseudo labels for unlabeled data, from which the student will learn. The student's performance on labeled data will be used as reward to train the teacher with *policy gradient*.

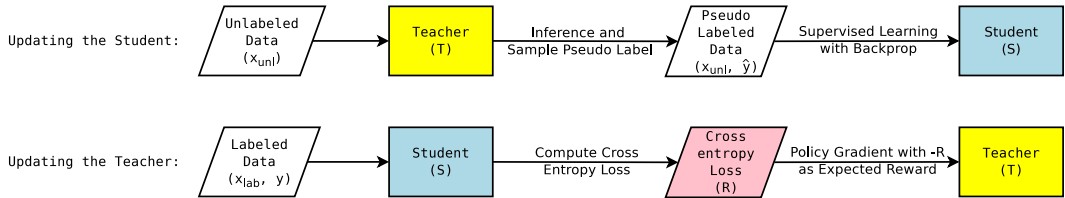

**Figure 1:** Each step of gradient descent in Coaching consists of two steps. **Updating the Student (top):** The teacher network $T$ samples the labels $\hat{y}$ of unlabeled data $x_{\text{unl}}$ for the student $S$ to learn from. **Updating the Teacher (bottom):** The teacher updates itself using policy gradient to improve the student's performance on labeled data $x_{\text{lab}}$.

---

[1]Code will be made available up on the paper's acceptance.

Experiments show that our method achieves significant improvements over state-of-the-art semi-supervised learning baselines and can be up to $10\times$ more data efficient than supervised learning. For instance, with CIFAR-10, only using 4,000 labeled examples, a WideResNet-28-2 can be coached to 96.11% accuracy, outperforming the same model trained with 45,000 labeled examples which achieves 94.9%. Meanwhile, on ImageNet, using ResNet-50 with only 10% labeled examples, our method achieves 72.94% top-1 accuracy, outperforming all existing semi-supervised learning methods with the same amount of labeled data, and approaching the top-1 accuracy of 76.3% of the same ResNet-50 trained with all labels. Coaching also scales to the high data regime. In particular, with all 1.28 million labeled examples from ImageNet, plus 9 million unlabeled and potentially out-of-distribution data from OpenImages (Kuznetsova et al., 2018), a ResNet-50 can be coached to the accuracy of 82.34%, which is a new state-of-the-art for the architecture without using extra labeled data.

## 2 METHOD

**Notations.** Let $T$, $S$ respectively be the teacher network and the student network in Coaching, and $\theta_T$, $\theta_S$ be their corresponding parameters. Since we work with both labeled data and unlabeled data, we use $(x_{\text{lab}}, y_{\text{lab}})$ to refer to a pair of an input and its corresponding label, and use $x_{\text{unl}}$ to refer to an unlabeled example. In addition, we use $\ell(x, y; \theta)$ to denote the cross entropy loss computed on input $x$ by with parameter $\theta$ on label $y$.

As shown in Figure 1, each training step in Coaching consists of two phases:

**Phase 1: The student learns from data pseudo labeled by the teacher.** In this phase, the teacher $T$ first performs a forward pass on $x_{\text{unl}}$ to compute the class distribution $P(\cdot|x_{\text{unl}}; \theta_T)$. From this distribution, the teacher samples a pseudo label $\hat{y}_{\text{unl}} \sim P(\cdot|x_{\text{unl}}; \theta_T)$. The pair $x_{\text{unl}}, \hat{y}_{\text{unl}}$ is then shown to the student $S$ to make an update on its parameters $\theta_S$. The update is based on the gradient computed by back-propagating from the cross entropy loss. For instance, if $\theta_S$ is updated using SGD, then:

$$\theta_S^{(t+1)} := \theta_S^t - \eta \cdot \underbrace{\left.\frac{\partial \ell(x_{\text{unl}}, \hat{y}_{\text{unl}}; \theta_S)}{\partial \theta_S}\right|_{\theta_S = \theta_S^{(t)}}}_{\triangleq\, g_S^{(t)}} = \theta_S^{(t)} - \eta \cdot g_S^{(t)}, \tag{1}$$

where $\eta$ is the learning rate.

**Phase 2: The teacher learns from the student's loss.** After the student updates its parameters as in Equation 1, its parameters $\theta_S^{(t+1)}$ is evaluated on a labeled example $x_{\text{lab}}, y_{\text{lab}}$ using the cross entropy loss. The goal of the teacher in Coaching is to give the pseudo labels $\hat{y}_{\text{unl}}$ such that *if the student is updated as in Equation 1, then the cross entropy loss $\ell(x_{\text{lab}}, y_{\text{lab}}; \theta_S^{(t+1)})$ will be minimized.*

Clearly, $\ell(x_{\text{lab}}, y_{\text{lab}}; \theta_S^{(t+1)})$ depends on $\theta_S^{(t+1)}$, which in turn depends on the pseudo label $\hat{y}_{\text{unl}}$ that the teacher samples. From the perspective of reinforcement learning, $\hat{y}_{\text{unl}}$ can be treated as an on-policy action of the teacher, which leads to the reward of $-\ell(x_{\text{lab}}, y_{\text{lab}}; \theta_S^{(t+1)})$. In this perspective, we propose to train $\theta_T$ to minimize the value of $\ell(x_{\text{lab}}, y_{\text{lab}}; \bar{\theta}_S^{(t+1)})$, where $\bar{\theta}_S^{(t+1)}$ is the expected destination that the teacher will guide the student to. This expectation is taken over all possible pseudo labels $\hat{y}_{\text{unl}}$. Formally,

$$\theta_T^* = \arg\min_{\theta_T} R(\theta_T) \text{ where } R(\theta_T) = \ell\left(x_{\text{lab}}, y_{\text{lab}}; \mathbb{E}_{\hat{y}_{\text{unl}} \sim P(\cdot|x_{\text{lab}}; \theta_T)}\left[\theta_S^{(t+1)}\right]\right) \tag{2}$$

To find $\theta_T^*$, we differentiate $R(\theta_T)$ in Equation 2 with respect to $\theta_T$. Here, we present the resulting gradient $g_T^{(t)}$, which has the form

$$g_T^{(t)} \approx \eta \cdot \underbrace{\left[\left(g_S^{(t)}\right)^\top \cdot \left(\left.\frac{\partial \ell(x_{\text{lab}}, y_{\text{lab}}; \theta_S)}{\partial \theta_S}\right|_{\theta_S = \theta_S^{(t+1)}}\right)^\top\right]}_{\text{a scalar } h^{(t)}} \cdot \left(\left.\frac{\partial \ell(x_{\text{unl}}, \hat{y}_{\text{unl}}; \theta_T)}{\partial \theta_T}\right|_{\theta_T = \theta_T^{(t)}}\right) \tag{3}$$

The full derivation can be found in Appendix A, but intuitively, the differentiation depends on two tools. The first tool is the is the chain rule, which we leverage to differentiate $R(\theta_T)$ with respect to $\theta_T$. The second tool is the REINFORCE equation (Williams, 1992), which we leverage to establish the relationship between $\mathbb{E}_{\hat{y}_{\mathrm{unl}}}\left[\theta_S^{(t+1)}\right]$ and $\theta_T$.

Coaching combines the two steps above in an SGD step. We summarize the method in Algorithm 1.

---

**Algorithm 1** The Coaching method.

---
**Input** : Labeled data $x_{\mathrm{lab}}, y_{\mathrm{lab}}$ and unlabeled data $x_{\mathrm{unl}}$.

1   Initialize $\theta_T^{(0)}$ and $\theta_S^{(0)}$

2   **for** $t = 0$ **to** $N - 1$ **do**

3      Sample an unlabeled example $x_{\mathrm{unl}}$ and a labeled example $x_{\mathrm{lab}}, y_{\mathrm{lab}}$

4      Sample $\hat{y}_{\mathrm{unl}} \sim P(\cdot | x_{\mathrm{unl}}; \theta_T)$

5      $\theta_S^{(t+1)} := \theta_S^{(t)} - \eta \cdot g_S^{(t)}$        ▷ *Compute $g_S^{(t)}$ with pseudo labels as in Equation 1 and update $\theta_S$*

6      $\theta_T^{(t+1)} := \theta_T^{(t)} - \eta \cdot h^{(t)} \cdot g_T^{(t)}$        ▷ *Compute the gradient $g_T^{(t)}$ as in Equation 3 and update $\theta_T$*

7   **end**

8   **return** $\theta_S^{(N)}$        ▷ *Only the student model is used for predictions and evaluations*

---

**Generalize to an arbitrary batch size.**   Above, we have only discussed Coaching for a single unlabeled data $x_{\mathrm{unl}}$ and a single labeled data $x_{\mathrm{lab}}, y_{\mathrm{lab}}$. Now, we describe how to scale Coaching to an arbitrary batch size. Scaling $x_{\mathrm{lab}}, y_{\mathrm{lab}}$ to a minibatch of labeled example, $X_{\mathrm{lab}}, Y_{\mathrm{lab}}$ is straightforward, as we can simply replace all computations of the cross entropy $\ell(x_{\mathrm{lab}}, y_{\mathrm{lab}}; \theta_S^{(t+1)})$ with the average cross entropy on the minibatch $\ell(X_{\mathrm{lab}}, Y_{\mathrm{lab}}; \theta_S^{(t+1)})$. To scale a single unlabeled example $x_{\mathrm{unl}}$ to a minibatch of unlabeled examples $X_{\mathrm{unl}} = \{x_{\mathrm{unl}}^{(1)}, x_{\mathrm{unl}}^{(2)}, ..., x_{\mathrm{unl}}^{(B)}\}$, we treat each batch of pseudo labels $\hat{Y}_{\mathrm{unl}}$ as a compound action sampled from the joint distribution

$$P\left(\hat{Y}_{\mathrm{unl}} \Big| X_{\mathrm{unl}}; \theta_T\right) = P\left(\hat{y}_{\mathrm{unl}}^{(1)}, \hat{y}_{\mathrm{unl}}^{(2)}, ..., \hat{y}_{\mathrm{unl}}^{(B)} \Big| X_{\mathrm{unl}}; \theta_T\right) = \prod_{i=1}^{B} P\left(\hat{y}_{\mathrm{unl}}^{(i)} \Big| x_{\mathrm{unl}}^{(i)}; \theta_T\right) \quad (4)$$

Since every pseudo label $\hat{y}_{\mathrm{unl}}^{(i)}$ is sampled independently, applying REINFORCE as in Equation 3 simply factors the per-instance cross entropy $\ell(x_{\mathrm{unl}}, \hat{y}_{\mathrm{unl}}; \theta_T)$ into the batch cross entropy $\sum_{i=1}^{B} \ell(x_{\mathrm{unl}}^{(i)}, \hat{y}_{\mathrm{unl}}^{(i)}; \theta_T)$.

## 3   EXPERIMENTS

Compared to other methods that use both labeled data and unlabeled data, Coaching has three main advantages:

1. The teacher does not only demonstrate its knowledge to the student but also adjusts its teaching strategy in an adaptive manner with the student, throughout the course of the student's learning.

2. The teacher in Coaching can benefit from advanced SSL techniques such as consistency regularization.

3. The student in Coaching *never learns directly from labeled data*. This does not only prevent overfitting when limited labeled data is available, but also allows us to finetune the trained student in Coaching directly on labeled data to further boost the student's performance.

We perform experiments to verify the strength of Coaching. In Section 3.1, we consider the low data regime with typical benchmarks for SSL methods. After that, in Section 3.2, we consider the high data regime which contains potentially out-of-distribution data.

**Model Architectures.**   In our experiments, our teacher model and our student model always have the same architecture but with different weights. For CIFAR-10 and SVHN, we use the WideResNet-28-2 (Zagoruyko & Komodakis, 2016), which has 1.45 million parameters. For ImageNet, we use a ResNet-50 (He et al., 2016), which has 25.5 million parameters. For experiments that train only one

model, we apply exponential moving average with a decay rate of $0.99$ on the weights of the model. For experiments that have a teacher model and a student model, we apply this exponential moving average on the weights of the student model only.

**Additional Implementation Details.** To improve the stability and accuracy of the method, we apply a few minor enhancements to *the teacher*:

1. *Use cosine distance instead of dot product.* As the dot product $h^{(t)}$ in Equation 3 has a large value range, in order to stabilize training, we compute $h^{(t)}$ using the gradients' cosine distance.

2. *Use a baseline for $h^{(t)}$.* To further reduce the variance of $h^{(t)}$, we maintain a moving average $b$ of $h^{(t)}$ and subtract $b$ from $h^{(t)}$ every time we compute $g_T^{(t)}$ as in Equation 3.

3. *Additional supervised loss for the teacher.* We find that adding the supervised loss $\ell(x_{\text{lab}}, y_{\text{lab}}; \theta_T)$ to the teacher's objective results in a faster learning and better student.

4. *Consistently regularize the teacher.* In the low data regime, consistency regularization improves the teacher and the student. More details are in Section 3.1.

5. *Pre-training the teacher.* When the number of classes is large, it is beneficial to initialize the teacher with a trained model so that the pseudo labels are better than random at the beginning of the student's learning. If we pre-train the teacher, Point 3 has minimal effect.

6. *Finetuning the student.* Since the student in Coaching only learns from unlabeled data and pseudo labels generated by the teacher, finetuning a converged student on labeled data often improves the student's performance.

The details mentioned above are mutually orthogonal. Since (1) and (2) are crucial to stabilize the Coaching process, they are always used in our experiments. In addition, we apply (3) and (4) to the low data regime, and apply (5) for the high data regime for computational efficiency and strong performance. We will explain these decisions in the corresponding sections.

## 3.1 Results on Low Data Regime

**Datasets.** We consider three datasets with reduced numbers of labeled instances: CIFAR-10 (Krizhevsky, 2009) with 4,000 labeled examples, SVHN (Netzer et al., 2011) with 1,000 labeled examples, and ImageNet (Russakovsky et al., 2015) with 128,000 labeled examples, which is approximately $10\%$ of the whole ImageNet. All images in these datasets are used as unlabeled examples, which means that even the labeled images can be used as unlabeled examples. We use the image size of $32 \times 32$ for CIFAR-10 and SVHN, and the image size of $224 \times 224$ for ImageNet. These datasets, label reductions, and image sizes are standard for low data image classification.

**Baselines.** We compare Coaching against 3 baseline training algorithms Purely Supervised, Pseudo-Label (Lee, 2013), and Unsupervised Data Augmentation (UDA; Xie et al. (2019)). We discuss these baselines more in Section 4. We choose these baselines for three reasons. First, the purely supervised baseline serves to verify our implementation and to demonstrate the overfitting of our models when labeled data is scarce. Second, comparing Coaching with Pseudo-Label confirms the benefits of continuing to train the teacher throughout the course of the student's learning. Finally, we compare against UDA because is the state-of-the-art on the datasets that we consider.

To ensure a fair comparison, we re-implement these baselines in our environment. We follow Oliver et al. (2018)'s train/eval/test splitting, and we use the same amount of resources to tune hyperparameters for our baselines as well as for Coaching. More details are in Appendix C.

**Additional baselines.** In addition to the three main baselines discussed above, we also include four other baselines: Temporal Ensemble (Laine & Aila, 2017), Mean Teacher (Tarvainen & Valpola, 2017), VAT (Miyato et al., 2018), LGA (Jackson & Schulman, 2019), ICT (Verma et al., 2019), and MixMatch (Berthelot et al., 2019). We use results reported by Oliver et al. (2018). Since these methods do not share the same controlled environment, the comparison to them is not direct, and should be contextualized as suggested by Oliver et al. (2018).

**Data augmentations.** In our implementation of UDA and Coaching, we use RandomAugment, which is a randomized augmentation strategy over all the operations in the search space of AutoAugment (Cubuk et al., 2019). We use RandomAugment because it is simple to implement, requires no expensive search, and achieves similar performance compared to UDA with AutoAugment. More details of RandomAugment can be found in Appendix C.2.

| Methods | CIFAR-10 (4,000) | SVHN (1,000) | ImageNet (10%) |
|---|---|---|---|
| Purely Supervised on full dataset | $94.92 \pm 0.17$ | $97.41 \pm 0.16$ | $76.89/93.27$ |
| Temporal Ensemble | $83.63 \pm 0.63$ | $92.81 \pm 0.27$ | $-$ |
| Mean Teacher | $84.13 \pm 0.28$ | $94.35 \pm 0.47$ | $-$ |
| VAT + EntMin | $86.87 \pm 0.39$ | $94.65 \pm 0.19$ | $-/83.39$ |
| LGA + VAT | $87.94 \pm 0.19$ | $93.42 \pm 0.36$ | $-$ |
| ICT | $92.71 \pm 0.02$ | $96.11 \pm 0.04$ | $-$ |
| MixMatch | $93.76 \pm 0.06$ | $96.73 \pm 0.31$ | $-$ |
| Purely Supervised | $82.14 \pm 0.25$ | $88.17 \pm 0.47$ | $57.75/80.23$ |
| Pseudo Labels | $83.79 \pm 0.11$ | $89.81 \pm 0.41$ | $58.21/82.19$ |
| UDA (our implementation) | $94.53 \pm 0.18$ | $97.11 \pm 0.17$ | $68.07/88.19$ |
| Coaching | $95.60 \pm 0.19$ | $97.79 \pm 0.11$ | $72.39/90.52$ |
| Coaching + Finetune | $\mathbf{96.11 \pm 0.07}$ | $\mathbf{98.01 \pm 0.07}$ | $\mathbf{72.94/90.80}$ |

**Table 1:** Image Classification Accuracy on reduced CIFAR-10, SVHN, and ImageNet. Higher is better. For CIFAR-10 and SVHN, we report mean $\pm$ std over 10 runs, while for ImageNet, we report Top-1/Top-5 accuracy of a single run. Results in the second block are taken from past papers, while the rest shares the same environment and hyper-parameter settings. All methods share the same model architecture: WideResNet-28-2 for CIFAR-10 and SVHN, and ResNet-50 for ImageNet.

**Main results.** In Table 1, we present our main results before and after finetuning the student on labeled data. The results confirm that Coaching significantly outperforms UDA and other strong baselines in semi-supervised learning.

On CIFAR-10 and SVHN, compared to the state-of-the-art UDA, Coaching's *error rate reduction* are roughly 30% and 10%. As UDA's accuracy is already relatively high, such error reductions are significant. On CIFAR-10, Coaching is also the first approach to exceed supervised learning on the all labels by using merely 4,000 labeled examples. Meanwhile, on ImageNet-10%, Coaching outperforms UDA by almost 5% in top-1 accuracy, going from 68.07% to 72.94%. Even prior to finetuning on labeled data, Coaching still outperforms UDA and other baselines.

**Comparing to existing state-of-the-art methods.** To the best of our knowledge, Coaching has achieved new state-of-the-art performances *among the same model architectures* on three datasets considered in this section.

For CIFAR-10 and SVHN, all existing better results use a larger model and more advanced regularization techniques. For instance, Xie et al. (2019) reports 97.3% with UDA (Xie et al., 2019), but their backbone model is PyramidNet, which has $18\times$ more parameters than WideResNet-28-2 and they train with Shake-Drop regularization (Yamada et al., 2018). Similarly, for ImageNet-10%, the only better published result is 73.21% top-1 accuracy, achieved by MOAM-$S^4L$ (Zhai et al., 2019). This accuracy is only slightly better than Coaching's 72.94%, but uses a $4\times$ wider ResNet-50. We believe that the enhancements in architectures, regularization techniques, and model sizes, can be applied to Coaching to further improve our results.

## 3.2 RESULTS ON HIGH DATA REGIME

We have seen Coaching achieves strong performance for low data image classification tasks. Another aspect of these tasks is that the unlabeled data also come from the same domain as the labeled data, which is a restricted assumption. In this section, we show that Coaching also excels in the regime where we have a large labeled dataset and an order of magnitude more unlabeled data. In this regime, we also test the performance of our method when the unlabeled set may have out-of-domain images, *i.e.,* the images belong to categories that do not exist in ImageNet.

**Datasets.** We experiment with all labeled examples in ImageNet. Additionally, we take unlabeled images from the entire $4^{\text{th}}$ version of OpenImages dataset (Kuznetsova et al., 2018), which has 9 million natural images. A few samples from OpenImages can be found in Figure 2. Unless otherwise specified, for both datasets, we use the image size of $224 \times 224$.

**Baselines.** Since this regime of high data has not been extensively studied, we are only aware of two relevant, strong baselines. Our first baseline is Billion-scale Semi-supervised Learning (Billion-scale SSL; Yalniz et al. (2019)). Billion-scale SSL uses unlabeled data from the YFCC100M dataset (Thomee et al., 2015), studies several self-training settings, with various model architectures for teachers and students. Here, we restrict our comparison to the settings that use ResNet-50 for both the teacher and the student. Our second baseline is UDA (Xie et al., 2019), for which the authors select unlabeled images algorithmically from the JFT dataset.[2]

Other than these baselines, we compare Coaching to techniques that enhance supervised learning, such as DropBlock (Ghiasi et al., 2018), CutMix (Yun et al., 2019), and FixRes (Touvron et al., 2019).

**Implementation details.** We implement Coaching the same as in Section 3.1, except for one part: Instead of directly training and consistently regularizing the teacher, we initialize the teacher using a pre-trained ResNet-50 (pre-trained on full ImageNet). Then, throughout the course of the student's learning, we only train the teacher to minimize the student's cross entropy loss. We do not use additional supervised loss for the teacher because because once the teacher is pre-trained, adding another loss to the teacher has minimal effect. We do not consistently regularize the teacher because Xie et al. (2019) has found that consistency regularization requires in-domain data, while we do not filter our unlabeled images from OpenImages.

| Methods | Unlabeled images | Image size | | Top-1 | Top-5 |
|---|---|---|---|---|---|
| | | Train | Test | | |
| Supervised | None | 224 | 224 | 76.89 | 93.27 |
| DropBlock | None | 224 | 224 | 78.35 | 94.15 |
| FixRes + CutMix | None | 224 | 320 | 79.8 | 94.9 |
| Coaching | OpenImages | 224 | 320 | 79.80 | 94.87 |
| FixRes | None | 224 | 384 | 79.1 | 94.6 |
| Coaching | OpenImages | 224 | 384 | **80.10** | **95.07** |
| Billion-scale SSL | YFCC 100M | 224 | 224 | 77.6 | − |
| Coaching | OpenImages | 224 | 224 | **78.62** | **94.26** |
| UDA | JFT | 331 | 331 | 79.04 | 94.45 |
| Coaching | OpenImages | 224 | 331 | 79.86 | 94.92 |
| Coaching+iterative | OpenImages | 224 | 331 | **82.34** | **96.09** |

**Table 2:** Image classification accuracy with full ImageNet plus unlabeled images. Results are organized by image size because image size has a strong impact on models' performance.

**Results.** We present our results in Table 2. As can be seen, Coaching outperforms all relevant SSL baselines. Specifically, for the image size of 224, Coaching outperforms Billion-scale SSL by about 1% top-1 accuracy, even though Billion-scale SSL uses 10 times more unlabeled data. Meanwhile, for the image size of 331, Coaching achieves the top-1 accuracy of 79.86%, comfortably outperforming the top-1 accuracy of 79.04% by UDA. This improvement is particularly significant, since Coaching simply uses all data from OpenImages, while UDA has to select and balance the class distribution of their unlabeled data using a pre-trained teacher. This difference suggests that the teacher in Coaching can give helpful pseudo labels to the student, even on potentially out-of-distribution data.

It is worth mentioning that Coaching also outperforms the strong supervised baselines of DropBlock and FixRes, and is on par with FixRes+CutMix. However, DropBlock and CutMix are both regularization techniques orthogonal to Coaching. Similar to consistency regularization in Section 3.1, these techniques can be incorporated into the teacher in Coaching to improve performance.

---

[2]Joint Foto Tree. A proprietary image dataset of Google, which is not available to public

**Comparing to state-of-the-art SSL results.** Yalniz et al. (2019) reports the top-1 accuracy of 81.2% for a ResNet-50 student. However, they need to pre-train a much bigger network ResNext-101-32x48 teacher (829 million parameters, 32x larger than ResNet-50) on 1 billion Instagram images with weak labels (Mahajan et al., 2018). Then, they use the pseudo-labels from this teacher to train a ResNet-50 student for 2 billion steps. The fact that they use weakly labeled data from Instagram, much bigger architecture in ResNext-101-32x48 makes their results not directly comparable to ours.

Meanwhile, without the need of a much bigger dataset and architecture as used in Yalniz et al. (2019), Coaching achieves almost as good top-1 accuracy. To achieve this, we iterate the process of Coaching by turning the student into the teacher after convergence. After 17 iterations, our final student achieves 82.34% top-1 accuracy on ImageNet, outperforming Yalniz et al. (2019)'s 81.2%, even though we do not have the weakly labeled data from Instagram.

**Insights about Coaching on OpenImages.** Figure 2 shows five images taken from OpenImages, along with their OpenImages tags and the top 5 classes predicted by a teacher trained on ImageNet. From the figure, we can see that there are non-trivial overlapping contents between the OpenImages tags and the ImageNet top classes, such as *sunglasses* in the first image. We also see that for the images whose contents match stronger with an ImageNet class, such as the first and the third image, the entropy of the teacher's prediction is smaller. As a result, when the teacher samples a pseudo label from these distribution, contents similar to an ImageNet class will receive more consistent labels, while content alien to ImageNet will have higher entropy on their labels. We suspect this is why a teacher trained on ImageNet can teach a student via pseudo labels on OpenImages.

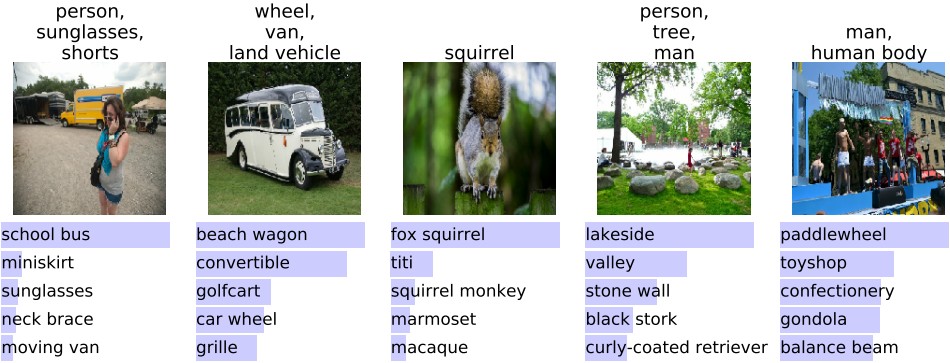

**Figure 2:** An illustration of why OpenImages help ImageNet classification. **Top:** OpenImages tags. **Middle:** A sample image from OpenImages. **Bottom:** Top 5 labels for the image predicted by a teacher ResNet-50 trained on ImageNet. Some OpenImages tags overlap significantly with some ImageNet classes, such as *wheel* and *car wheel* in the second image. The class predictions also have a higher entropy when the ImageNet classes overlap less with the OpenImages contents (images 2, 4, 5), than when the ImageNet classes overlap more (images 1, 3).

### 3.3 ANALYSIS

**Ablation Study of Implementation Details.** To understand the contribution of each implementation detail of Coaching, we study their contributions on top of a purely supervised model. We conduct this study on ImageNet-10% and visualize the results in Figure 3. From the figure we see that RandomAugment and UDA both improve the final accuracy significantly, respectively by 3.13% and 7.19% top-1 accuracy. On top of UDA, Coaching delivers a smaller improvement of 4.32% top-1 accuracy. However, since UDA's accuracy is already high, we believe that the improvement of 4.32% top-1 accuracy is significant. Finally, finetuning only slightly improves over Coaching. However, this extra boost is a unique advantage of Coaching: it is possible for the student in Coaching to finetune on labeled data because the student never directly learns from these labeled data.

**Coaching overfits less than Supervised Learning.** In our Coaching framework, the student never directly learns from labeled data. This behavior is helps the student to avoid overfitting, especially when labeled data is scarce. In Figure 4, we visualize the training accuracy of Coaching and Supervised Learning on CIFAR-10 with 4,000 labels and on ImageNet with 10% labels. As shown, the training accuracy of both the teacher and the student of Coaching stay relatively low. Meanwhile, the training accuracy of the supervised model eventually reaches 100% and causes overfitting.

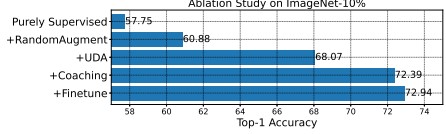

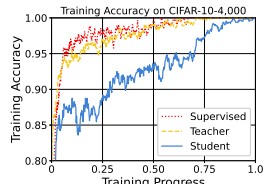
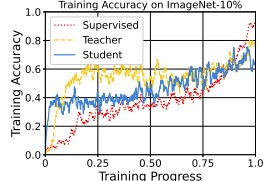

**Figure 3:** Breakdown of the gains of different components in Coaching. The gain of Coaching over UDA, albeit smaller than the gain of UDA over RandomAugment, is significant as UDA is already very strong.

**Figure 4:** Training accuracy of Coaching and of supervised learning on CIFAR-10-4,000 and ImageNet-10%. Both the teacher and the student in Coaching have lower training accuracy, effectively avoiding overfitting.

## 4 RELATED WORK

**Pseudo-Label.** Pseudo-Label (Lee, 2013) is one of the simplest semi-supervised learning algorithms: First, a teacher model is trained on labeled data. Then, the converged teacher model generates pseudo labels for unlabeled data. These unlabeled data and their pseudo labels are combined with the labeled data to train another model, which is called the student model. An inherent weakness of Pseudo-Label is that once the teacher generates an incorrect pseudo label for an unlabeled datum, the student can only naively learn from this wrong label. This phenomenon is called the confirmation bias. Arazo et al. (2019) addressed the confirmation bias by generating soft labels from the teacher and by adding noise to these labels. However, this is a manual fix from an outside model designer. The main difference between Pseudo-Label and Coaching is that in Coaching, the teacher is trained along with the student throughout the course of training. This allows wrong knowledge learned by the teacher to be fixed in an end-to-end manner, leading to stronger performances.

**Semi-supervised Learning (SSL).** Pseudo-Label belongs to a more general group of algorithms known as Semi-supervised Learning. Unlike Pseudo-Label, typical SSL methods combine both labeled and unlabeled data to train *a single model*. Hence, the objective function of SSL is typically the sum of a supervised loss and an unsupervised loss. The supervised loss is often the cross-entropy computed on the labeled data. Meanwhile, the unsupervised loss can be a self-supervised loss (Rasmus et al., 2015; Noroozi & Favaro, 2018; Gidaris et al., 2018), or consistency regularization (Laine & Aila, 2017; Tarvainen & Valpola, 2017; Miyato et al., 2018; Berthelot et al., 2019; Xie et al., 2019). Self-supervised losses typically encourage the model to develop a common sense about the images. Meanwhile, consistency regularization enforces that the model is invariant against certain transformations of the data. The main difference between Coaching and SSL methods is that the student in Coaching never learns directly from labeled data. This helps the student in Coaching to avoid overfitting to labeled data, especially when labeled data is limited.

**Meta Learning.** In Meta Learning, there is typically an outer loop that optimizes the performance of a model trained in an inner loop (Finn et al., 2017; Metz et al., 2019). Meta Learning has been applied to perform self-training and SSL in the low data regime (Agarwal et al., 2019; Ren et al., 2018; Boney & Ilin, 2018; Hsu et al., 2019). A crucial difference between Coaching and Meta Learning is that in Coaching, the pseudo labels are chosen to improve the student, and hence there is no need for an outer loop. We suspect this is an advantage of our method, since gradients to be very powerful for models to navigate in the parameter space.

## 5 CONCLUSION

In this paper, we proposed the Coaching method for semi-supervised learning. Key to Coaching is the idea that the teacher learns from the student's loss and improves itself to generate pseudo labels in a way that helps student's learning the most. The learning process in Coaching consists of two main updates: updating the student based on the pseudo labeled data produced by the teacher and updating the teacher based on the student's performance. Experiments on standard CIFAR-10 and SVHN show that Coaching is much better than supervised learning and consistenly better than other semi-supervised learning methods. Coaching scales well to large problems, and successfully uses out-of-domain data to improve ImageNet classification.

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

# A   DERIVATION OF THE TEACHER'S UPDATE RULE

In this section, we present the detailed derivation of the Teacher's update rule in Equation 3 from Section 2.

**Mathematical Notations and Conventions.**   Since we will work with the chain rule, we use the standard Jacobian notations.[3] Specifically, for a differentiable function $f : \mathbb{R}^m \to \mathbb{R}^n$, and for a vector $x \in \mathbb{R}^m$, we use the notation $\frac{\partial f}{\partial x} \in \mathbb{R}^{n \times m}$ to denote *the Jacobian matrix* of $f$, whose dimension is $n \times m$. Additionally, when we mention the Jacobian of a function $f$ at multiple points such as $x_1$ and $x_2$, we will use the notations of $\left. \frac{\partial f}{\partial x} \right|_{x=x_1}$ and $\left. \frac{\partial f}{\partial x} \right|_{x=x_2}$.

Furthermore, by mathematical conventions, a vector $v \in \mathbb{R}^n$ is treated as a *column matrix* – that is, a matrix of size $n \times 1$. For this reason, the gradient vector of a multi-variable real-valued function is actually the transpose of of its Jacobian matrix.

Finally, all multiplications in this section are standard matrix multiplications. If an operand is a vector, then as discussed in the previous paragraph, the operand is treated as a column matrix.

**Dimension Annotations.**   Understanding that these notations and conventions might cause confusions, in the derivation below, we annotate the dimensions of the computed quantities to ensure that there is no confusion caused to our readers. To this end, we respectively use $|S|$ and $|T|$ to denote the dimensions of the parameters $\theta_S$, $\theta_T$. That is, $\theta_S \in \mathbb{R}^{|S| \times 1}$ and $\theta_T \in \mathbb{R}^{|T| \times 1}$.

We now present the derivation. We need to compute:

$$\underbrace{\frac{\partial R}{\partial \theta_T}}_{1 \times |T|} = \frac{\partial}{\partial \theta_T} \ell \left( x_{\text{lab}}, y_{\text{lab}}; \mathbb{E}_{\hat{y}_{\text{unl}} \sim P(\cdot | x_{\text{unl}}; \theta_T)} \left[ \theta_S^{(t+1)} \right] \right) \tag{5}$$

To simplify our notation, let us define

$$\underbrace{\bar{\theta}_S^{(t+1)}}_{|S| \times 1} \triangleq \mathbb{E}_{\hat{y}_{\text{unl}} \sim P(\cdot | x_{\text{unl}}; \theta_T)} \left[ \theta_S^{(t+1)} \right] \tag{6}$$

Then, by the chain rule, we have

$$
\begin{aligned}
\underbrace{\frac{\partial R}{\partial \theta_T}}_{1 \times |T|} &= \frac{\partial}{\partial \theta_T} \ell \left( x_{\text{lab}}, y_{\text{lab}}; \mathbb{E}_{\hat{y}_{\text{unl}} \sim P(\cdot | x_{\text{unl}}; \theta_T)} \left[ \theta_S^{(t+1)} \right] \right) \\
&= \frac{\partial}{\partial \theta_T} \ell \left( x_{\text{lab}}, y_{\text{lab}}; \bar{\theta}_S^{(t+1)} \right) \\
&= \underbrace{\left. \frac{\partial \ell \left( x_{\text{lab}}, y_{\text{lab}}; \theta_S \right)}{\partial \theta_S} \right|_{\theta_S = \bar{\theta}_S^{(t+1)}}}_{1 \times |S|} \cdot \underbrace{\frac{\partial \bar{\theta}_S^{(t+1)}}{\partial \theta_T}}_{|S| \times |T|}
\end{aligned}
\tag{7}
$$

The first factor in Equation 7 can be simply computed via back-propagation. We now focus on the second term. We have

$$
\begin{aligned}
\underbrace{\frac{\partial \bar{\theta}_S^{(t+1)}}{\partial \theta_T}}_{|S| \times |T|} &= \frac{\partial}{\partial \theta_T} \mathbb{E}_{\hat{y}_{\text{unl}} \sim P(\cdot | x_{\text{unl}}; \theta_T)} \left[ \theta_S^{(t+1)} \right] \\
&= \frac{\partial}{\partial \theta_T} \mathbb{E}_{\hat{y}_{\text{unl}} \sim P(\cdot | x_{\text{unl}}; \theta_T)} \left[ \theta_S^{(t)} - \eta \cdot \left( \left. \frac{\partial \ell \left( x_{\text{unl}}, \hat{y}_{\text{unl}}; \theta_S \right)}{\partial \theta_S} \right|_{\theta_S = \theta_S^{(t)}} \right)^\top \right]
\end{aligned}
\tag{8}
$$

---

[3]Standard: https://en.wikipedia.org/wiki/Jacobian_matrix_and_determinant

Note that in Equation 8 above, the Jacobian of $\ell(x_{\mathrm{unl}}, \hat{y}_{\mathrm{unl}}; \theta_S)$, which has dimension $1 \times |S|$, needs to be transposed to match the dimension of $\theta_S^{(t)}$, which, as we discussed above, conventionally has dimension $|S| \times 1$.

Now, since $\theta_S^{(t)}$ in Equation 8 does not depend on $\theta_T$, we can leave it out of subsequent derivations. Also, to simplify notations, let us define *the gradient*

$$
\underbrace{g_S^{(t)}(\hat{y}_{\mathrm{unl}})}_{|S| \times |1|} \triangleq \left( \left. \frac{\partial \ell\left(x_{\mathrm{unl}}, \hat{y}_{\mathrm{unl}}; \theta_S\right)}{\partial \theta_S} \right|_{\theta_S = \theta_S^{(t)}} \right)^{\top} \tag{9}
$$

Then, Equation 8 becomes

$$
\underbrace{\frac{\partial \bar{\theta}_S^{(t+1)}}{\partial \theta_T}}_{|S| \times |T|} = -\eta \cdot \frac{\partial}{\partial \theta_T} \mathbb{E}_{\hat{y}_{\mathrm{unl}} \sim P(\cdot | x_{\mathrm{unl}}; \theta_T)} \Big[ \underbrace{g_S^{(t)}(\hat{y}_{\mathrm{unl}})}_{|S| \times 1} \Big] \tag{10}
$$

Since $g_S^{(t)}(\hat{y}_{\mathrm{unl}})$ has no dependency on on $\theta_T$, except for via $\hat{y}_{\mathrm{unl}}$, we can apply the REINFORCE equation (Williams, 1992) to achieve

$$
\begin{aligned}
\underbrace{\frac{\partial \bar{\theta}_S^{(t+1)}}{\partial \theta_T}}_{|S| \times |T|} &= -\eta \cdot \frac{\partial}{\partial \theta_T} \mathbb{E}_{\hat{y}_{\mathrm{unl}} \sim P(\cdot | x_{\mathrm{unl}}; \theta_T)} \Big[ g_S^{(t)}(\hat{y}_{\mathrm{unl}}) \Big] \\
&= -\eta \cdot \mathbb{E}_{\hat{y}_{\mathrm{unl}} \sim P(\cdot | x_{\mathrm{unl}}; \theta_T)} \Big[ \underbrace{g_S^{(t)}(\hat{y}_{\mathrm{unl}})}_{|S| \times 1} \cdot \underbrace{\frac{\partial \log P\left(\hat{y}_{\mathrm{unl}} | x_{\mathrm{unl}}; \theta_T\right)}{\partial \theta_T}}_{1 \times |T|} \Big] \\
&= \eta \cdot \mathbb{E}_{\hat{y}_{\mathrm{unl}} \sim P(\cdot | x_{\mathrm{unl}}; \theta_T)} \Big[ \underbrace{g_S^{(t)}(\hat{y}_{\mathrm{unl}})}_{|S| \times 1} \cdot \underbrace{\frac{\partial \ell\left(x_{\mathrm{unl}}, \hat{y}_{\mathrm{unl}}; \theta_T\right)}{\partial \theta_T}}_{1 \times |T|} \Big]
\end{aligned} \tag{11}
$$

Here, the last equality in Equation 11 is is due to the definition of the cross entropy loss, which is the negative of the log-prob term in the previous line.

Now, we can substitute Equation 11 into Equation 7 to obtain

$$
\begin{aligned}
\underbrace{\frac{\partial R}{\partial \theta_T}}_{1 \times |T|} &= \underbrace{\left. \frac{\partial \ell\left(x_{\mathrm{lab}}, y_{\mathrm{lab}}; \theta_S\right)}{\partial \theta_S} \right|_{\theta_S = \bar{\theta}_S^{(t+1)}}}_{1 \times |S|} \cdot \underbrace{\frac{\partial \bar{\theta}_S^{(t+1)}}{\partial \theta_T}}_{|S| \times |T|} \\
&= \eta \cdot \underbrace{\left. \frac{\partial \ell\left(x_{\mathrm{lab}}, y_{\mathrm{lab}}; \theta_S\right)}{\partial \theta_S} \right|_{\theta_S = \bar{\theta}_S^{(t+1)}}}_{1 \times |S|} \cdot \mathbb{E}_{\hat{y}_{\mathrm{unl}} \sim P(\cdot | x_{\mathrm{unl}}; \theta_T)} \Big[ \underbrace{g_S^{(t)}(\hat{y}_{\mathrm{unl}})}_{|S| \times 1} \cdot \underbrace{\frac{\partial \ell\left(x_{\mathrm{unl}}, \hat{y}_{\mathrm{unl}}; \theta_T\right)}{\partial \theta_T}}_{1 \times |T|} \Big]
\end{aligned} \tag{12}
$$

Finally, if we use Monte Carlo approximation for every term in Equation 12 using the sampled $\hat{y}_{\mathrm{unl}}$, then we have Equation 3 from Section 2. Note that in Section 2, we use the gradient notation, which results in the transposes.

## B  TRAINING SPEED

Coaching performs up to 5 forward passes and 3 backward passes. Compared to vanilla back-propagation training, this is more 3 than times more expensive in FLOPs. However, many computations in Coaching are parallelizable. For example, the forward pass of the student and the forward pass for the teacher on unlabeled data (the top half of Figure 1), can be run in parallel since they do not depend on each other. Therefore, on computing hardware with sufficient memory, we find Coaching to be between 2 and 2.5 times slower than standard back-propagation training.

## C  EXPERIMENTAL DETAILS

### C.1  DATASET SPLITS

We describe how we select the reduced datasets for the experiments on low data image classification in Section 3.1.

For CIFAR-10, we download the five training data batch files from `www.cs.toronto.edu/~kriz/cifar.html`. Then, we load all the images into a list of 50,000 images, keeping the order as downloaded. The fisrt 5,000 images ares reserved for validation. The next 4,000 images are used as labeled data. For SVHN, we download the data from the `mat` files on `ufldl.stanford.edu/housenumbers/`, and follow the same procedure as with CIFAR-10. We note that this selection process leads to a slight imbalance in the class distribution for both CIFAR-10 and SVHN, but the settings are the same for all of our experiments.

For ImageNet, we follow the procedure in `github.com/tensorflow/models/blob/master/research/inception/inception/data/download_and_preprocess_imagenet.sh`. This results in 1,024 training `TFRecord` shards of approximately the same size. The order of the images in these shards are deterministic. For ImageNet-10%, we use the first 102 shards; for ImageNet-20%, we use the first 204 shards; and so on. The last 20 shards, corresponding to roughly 25,000 images, are reserved for hyper-parameters tuning.

### C.2  RANDOMAUGMENT: A DATA AUGMENTATION POLICY

We develop a data augmentation policy that achieves similarly high performance with AutoAugment (Cubuk et al., 2019) in a few cases that we consider, but which does not require learning a controller to generate policies. We names our policy RandomAugment. Our goal when developing RandomAugment is not to outperform AutoAugment, which is why we do not conduct extensive experiments with RandomAugment. Instead, we simply want to avoid *indirectly* using labeled data for our experiments, especially for the low data regime experiments in Section 3.1.

Each policy of RandomAugment consists of two operations that applied sequentially on an image. Each operation applies a uniformly sampled transformation with probability 0.5, and with a level uniformly chosen between 1 and 10. For a more comprehensive discussion of the probability and the level of a transformation, we refer readers to the AutoAugment paper (Cubuk et al., 2019).

| **CIFAR-10** and **ImageNet** | **SVHN** |
|---|---|
| AutoContrast | AutoContrast |
| Brightness | Brightness |
| Color | Color |
| Contrast | Contrast |
| Equalize | Equalize |
| Invert | Invert |
| Sharpness | Sharpness |
| Posterize | Posterize |
| Sample Pairing | Solarize |
| Solarize | ShearX |
| Rotate | ShearY |
| ShearX | TranslateY |
| ShearY | |
| TranslateX | |
| TranslateY | |

**Table 3:** Transformations that RandomAugment uniformly samples for our datasets. We refer our readers to Cubuk et al. (2019) for the detailed descriptions of these transformations.

We manually design the set of transformations for each of our datasets: CIFAR-10, ImageNet, and SVHN. The set of transformations for CIFAR-10 and for ImageNet are the same, and are slightly different from the set of transformation for SVHN. This is because the numbers in the SVHN have a different requirement for invariant. For instance, numbers should *not* be invariant against rotations like 6 and 9, and should *not* be invariant against horizontal translation like 3 and 8. Table 3 presents the transformation for our dataset. In addition to these operations, we only allow RandomAugment to select the three transformations AutoContrast, Brightness, and Invert in the first augmenting

transformation. This is to avoid a few degenerating cases. For instance, when Brightness is applied twice on an image, both times with small levels, the image will become almost black.

In our experiments, RandomAugment's performance is not far behind compared to AutoAugment. For example, on full ImageNet with ResNet-50, RandomAugment achives 77.98% top-1 accuracy, which is close to the top-1 accuracy of 77.6% reported by Cubuk et al. (2019).

## C.3 HYPER-PARAMETERS

To tune hyper-parameters, we follow Oliver et al. (2018) and allow each method to have 128 trials of hyper-parameters. When we tune, we let each model train for up to 50,000 steps. The optimal hyper-parameters are then used to run experiments that last for much more steps, as we report below. In our experiments with Coaching, training for more steps typically leads to stronger results. We stop at 1 million steps for CIFAR-10 and SVHN, and at 0.5 million steps for ImageNet simply because otherwise, these experiments will take too long. Meanwhile, in our experiments with purely supervised learning, Pseudo-Labels, and UDA, training for more steps overfits the models, and we have to employ early stopping.

We report the hyper-parameters for our baselines and for Coaching in Section 3.1. For the high-resource experiments in Section 3.2, we use the same hyper-parameters, because tuning them is too expensive. Our hyper-parameters can be found in Table 4, 5, 6.

We note that our settings for UDA is different from originally reported by Xie et al. (2019). In their work, Xie et al. (2019) use a much larger batch size for their UDA objective. In our implementation of UDA, we keep these batch sizes the same. This leads to a much easier implementation of data parallelism in our framework, TensorFlow (Abadi et al., 2016) running on TPU big pods. To compensate for the difference, we train all UDA baselines for much longer than Xie et al. (2019). During the training process, we also mask out the supervised examples with high confidence. Effectively, our UDA model receives roughly the same amount of training with labeled examples and unlabeled examples as the models in Xie et al. (2019). We have also verified that on ImageNet-10% with the augmentation policy from AutoAugment (Cubuk et al., 2019), our UDA implementation achives 68.77% top-1 accuracy, which is similar to 68.66% that Xie et al. (2019) reported.

| Hyper-parameter | CIFAR-10 | SVHN | ImageNet |
|---|---|---|---|
| Weight decay | 0.0005 | 0.001 | 0.0002 |
| Label smoothing | 0 | 0 | 0.1 |
| Batch normalization decay | 0.99 | 0.99 | 0.99 |
| Learning rate | 0.4 | 0.05 | 1.28 |
| Number of training steps | 50,000 | 50,000 | 40,000 |
| Number of warm up steps | 2500 | 0 | 2000 |
| Batch size | 1024 | 128 | 2048 |
| Dropout rate | 0.4 | 0.5 | 0.2 |
| Pseudo label threshold | 0.95 | 0.975 | 0.7 |

**Table 4:** Hyper-parameters for supervised learning and Pseudo-Labels.

| Hyper-parameter | CIFAR-10 | SVHN | ImageNet |
|---|---|---|---|
| Weight decay | 0.0005 | 0.0005 | 0.0002 |
| Label smoothing | 0 | 0 | 0.1 |
| Batch normalization decay | 0.99 | 0.99 | 0.99 |
| Learning rate | 0.3 | 0.4 | 1.28 |
| Number of training steps | 1,000,000 | 1,000,000 | 500,000 |
| Number of warm up steps | 5,000 | 5,000 | 5,000 |
| Batch size | 128 | 128 | 2048 |
| Dropout rate | 0.5 | 0.6 | 0.25 |
| UDA factor | 2.5 | 1 | 20 |
| UDA temperature | 0.7 | 0.8 | 0.7 |

**Table 5:** Hyper-parameters for UDA. Unlike originally done by Xie et al. (2019), we do not use a larger batch size for the UDA objective. Instead, we use the same batch size for both the labeled objective and the unlabeled objective. This is to avoid instances where some particularly small batch sizes for the labeled objective cannot be split on our computational hardware.

| | Hyper-parameter | CIFAR-10 | SVHN | ImageNet |
|---|---|---|---|---|
| Common | Weight decay | 0.0005 | 0.0005 | 0.0002 |
| | Label smoothing | 0.1 | 0.1 | 0.1 |
| | Batch normalization decay | 0.99 | 0.99 | 0.99 |
| | Number of training steps | 1,000,000 | 1,000,000 | 500,000 |
| | Number of warm up steps | 2,000 | 2,000 | 1,000 |
| Student | Learning rate | 0.3 | 0.15 | 0.8 |
| | Batch size | 128 | 128 | 2048 |
| | Dropout rate | 0.35 | 0.45 | 0.1 |
| Teacher | Learning rate | 0.125 | 0.05 | 0.5 |
| | Batch size | 128 | 128 | 2048 |
| | Dropout rate | 0.5 | 0.65 | 0.1 |
| | UDA factor | 1.0 | 2.5 | 16.0 |
| | UDA temperature | 0.8 | 1.25 | 0.75 |

**Table 6:** Hyper-parameters for Coaching.

