# OpenReview forum: "Semi-supervised Learning by Coaching"
_ICLR.cc/2020/Conference — Reject_

### Official Review · AnonReviewer1 · 2019-10-22
**Official Blind Review #1**

**Rating:** 3

**Review:**

This paper provides a simple but novel coaching method for teacher-student based semi-supervised learning framework. The coaching method consists of a two-stage update: first update Student network according to the pseudo label produced by Teacher network on the unlabeled dataset; second update Teacher network according to the Student network's performance on the labeled dataset. The authors propose a novel policy gradient update for the Teacher network. The authors evaluate the coaching method on several different semi-supervised learning dataset CIFAR-10, SVHN and ImageNet. The comparison with baselines are thorough. I appreciate that the authors explain the design of the experiments and tuning in details.

I vote for acceptance but I still have some questions. I would be willing to increase my score if the authors address my questions in the rebuttal.
1. The motivation of coaching is not accurate. In the second sentence of Introduction, the authors mention "Although coaches do not play as well as the players". However, we are training neural networks. There is no such thing as "coaches do not play as well as the players". They are the same parametric neural networks. (The authors also use the same architectures for both teacher and student neural networks.) I would remove this analogy sentence.
2. The authors mention that their coaching method beats the fully supervised learning on the CIFAR-10. I am not convinced by this result. The author explained that this is due to the less overfitting in the student network. However, besides coaching, there are many other regularization method we can use to avoid overfitting. Using far less labels in training strictly reduces the amount of information we have. There is a simple test the authors can do. We can use the full labelled data set and use sampled results from Teacher network to train a Student network on it. We can perform coaching on this regime. This coaching on full dataset should do better than coaching on partially-labeled dataset (4000 labels).
3. The state of art for CIFAR-10 is 99% now [1]. It would be nice to see the performance of author's approach applies to the state-of-art network/structure.

References:
[1] Huang, Yanping, et al. "Gpipe: Efficient training of giant neural networks using pipeline parallelism." arXiv preprint arXiv:1811.06965 (2018).

-----
The authors' response does not answer the questions about motivation and why the method works, which is also questioned by the two other reviewers. For example, why the authors need to have two networks is unclear; I guess that true labels should be more helpful to guide a student network to learn than a teacher network. Even though the authors seem to have state-of-art semi-supervised learning results, some extra explanations are needed.

**Experience Assessment:**

I have read many papers in this area.

**Review Assessment: Checking Correctness Of Derivations And Theory:**

I carefully checked the derivations and theory.

**Review Assessment: Checking Correctness Of Experiments:**

I carefully checked the experiments.

**Review Assessment: Thoroughness In Paper Reading:**

I read the paper thoroughly.

---

> ### Author Response · Authors · 2019-11-06
> **Motivations for Coaching. Also, "beats the fully supervised learning" needs to be considered on the same architectures**
>
> Thank you for time and your comments.
>
> [On Coaching’s Motivations]
> We agree with your point here. We will consider rephrasing the analogy with coaching in sports in order to avoid causing misunderstandings to our readers.
>
> You are correct that our student network shares the same architecture with our teacher network. However, this only entails that these networks have the same *learning capacity*. In fact, when put into our framework, the teacher’s focus is *not* to learn anything for itself, but instead, to improve the student’s performance.
>
> In our experiments, it is indeed the case that the student typically has a higher accuracy than the teacher. For example, on ImageNet-10%, our teacher network (trained with UDA [1]) has a top-1 accuracy of 69.11%, which is around the same performance that [1] reported. Meanwhile, in that same experiment, the student achieves a much stronger top-1 accuracy of 73.32%.
>
> [On Coaching with Full CIFAR-10]
> We are not sure what you meant by this request. In order to performance Coaching, we need have both labeled data and unlabeled data.
>
> The setting of using 4,000 labeled examples from CIFAR-10 is designed to limit the amount of *labeled data* that a training algorithm can use. If we allow the teacher to see all the labels, as seemingly suggested by your “simple test”, then we would obviously have a stronger student. However, this is not the point of the experiments, which is to demonstrate the Coaching works *in the low-data regime*.
>
> [On CIFAR-10’s State-of-the-Art]
> To compare training algorithms, accuracies should be compared among the same model architectures, trained using the same amount of labeled data.
>
> To achieve 99% accuracy on CIFAR-10, the GPipe paper (Huang et al, 2018) that you mentioned uses a large version of AmoebaNet-C which has 600M parameters and hence, is much larger than our WideResNet-28-2 (1.45M parameters). Furthermore, the 99% accuracy is achieved by finetuning a model *pretrained on ImageNet*.
>
> In fact, with 4000 labeled examples, we have achieved the accuracy of 98.21% by applying Coaching to EfficientNet-B0 [2] (of course, without pretraining on ImageNet). This accuracy slightly outperforms the accuracy of 98.1% that [2] achieved by pretraining on ImageNet and then finetuning on full CIFAR-10. Note that EfficientNet-B0 only has 4M parameters, which is much smaller than 600M of AmoebaNet-C.
>
> [1] Unsupervised Data Augmentation for Consistency Training. Qizhe Xie, Zihang Dai, Eduard Hovy, Minh-Thang Luong, Quoc V. Le. https://arxiv.org/abs/1904.12848
>
> [2] EfficientNet: Rethinking Model Scaling for Convolutional Neural Networks. Mingxing Tan, Quoc V. Le. https://arxiv.org/abs/1905.11946

---

### Official Review · AnonReviewer2 · 2019-10-23
**Official Blind Review #2**

**Rating:** 3

**Review:**

This paper studies the teacher-student models in semi-supervised learning. Unlike previous methods in which only the student will learn from the teacher, this paper proposes a method to let the teacher learn from the student by reinforcement learning. Experimental results demonstrate the proposal’s performance.

The paper achieves some good empirical results compared to other baselines. However, the proposed method is implemented with many tricks listed on Page 4, and with data augmentation techniques, which may not be used in previous methods. Additionally, the paper is weak in technology. There is no clear explanation of why the proposed method works except a metaphor for sports coaches. I vote for a clear rejection of the paper.

First, the paper is weak in experiments. It works hard to achieve a good experimental result, through many tricks listed in the “Additional Implementation Details” in Page 4, and through the data augmentation used in Page 5. However, these tricks to improve the performance may not be used in previous methods, as the paper does not run experiments on baselines under the same setting, but use the results reported in Oliver et al. (2018). Additionally, the paper only uses one number of labeled data for each data set, it makes readers doubt that the proposed method only works under this number of labeled data.

The paper fails to clearly state why we need to let the teacher learn from the student. Actually, I doubt if this is necessary. Given the strong learning capacity of neural networks, the proposed method will easily be overfitting. Assume we have a very weak student network at the beginning, then by training in the way proposed in the paper, the teacher network will have all labeled data classified correct, and all unlabeled data classified into the same labels as the student network. I cannot see from the simple proposal why such overfitting can be avoided.

The paper is weak in both technology and experiments. It is also poorly written without clearly stating the motivation for this problem. I would vote for a reject for the paper.

-------------------------------------------
The rebuttal has cleared some of my concerns. However, it is still not clear why the proposed method work and how does it prevents overfitting. The paper also needs more experimental results to confirm its effectiveness. I will increase my score a little bit, but would not vote for an accept this time. But I believe with further revision, the paper may be worth publishing in the future, if the questions in all reviews can be addressed.

**Experience Assessment:**

I have read many papers in this area.

**Review Assessment: Checking Correctness Of Derivations And Theory:**

N/A

**Review Assessment: Checking Correctness Of Experiments:**

I assessed the sensibility of the experiments.

**Review Assessment: Thoroughness In Paper Reading:**

I read the paper at least twice and used my best judgement in assessing the paper.

---

> ### Author Response · Authors · 2019-11-06
> **We are dismayed by your rating. We try to address your concerns.**
>
> Thank you for time and your comments. By reading your review, we believe there are some points in our paper that you might be misunderstanding. Below, we try to clarify and address these points.
>
> [On the Lack of Explanation for Coaching’s Strong Performance]
> We respectfully disagree with you that we have “no clear explanation of why the proposed method works except a metaphor for sports coaches”. Specifically:
>
> We have mentioned in Section 3.3 on Page 7 of our paper that *one advantage* of Coaching is that it avoids overfitting. In fact, in many of our experiments, Coaching models are trained for 1 million steps (Tables 4, 5, 6; Appendix C).
>
> In the low-data regime, *no* previous method can train for that long without severely overfitting. For example, this screenshot ( https://pasteboard.co/IFkOBbz.png ) shows such a case for ImageNet-10% after just 50K steps.
>
> One can argue that a large dropout rate, a large weight-decay rate, as well as other regularization methods, will also avoid overfitting. However, they are all orthogonal to coaching, and *none* of those methods has ever achieved the strong performance as Coaching. Thus, at the very least, Coaching provides a form of very effective regularization.
>
> There could be other reasons for the strong performance of Coaching, and they are not just our Additional Implementation Details, as you incorrectly assumed.
>
> [On the Performance of Coaching without Additional Techniques]
> We also disagree with you that the strong performance of Coaching is associated to *just* the Additional Implementation Details.
>
> First, we have provided an evidence in Figure 3 on Page 8, which shows that Coaching+UDA improves over UDA by 4.32% top-1 accuracy on ImageNet, which is a very significant boost for this dataset. As UDA uses *all* techniques (consistency loss, data augmentation, etc.), the comparison between Coaching+UDA and UDA is a *controlled* experiment that demonstrates the strength of Coaching.
>
> Second, our use of cosine distance and a moving average baseline are all the features of the Coaching method. This is not a weakness of Coaching, just like the moving average baseline is not a weakness of *all* Reinforcement Learning algorithms that use policy gradient. To our knowledge, we do not reject policy gradient methods just because they use a baseline to improve their results. Instead, we simply follow such recipe.
>
> Third, to further address your concern about our Additional Implementation Techniques, we have also run Coaching on ImageNet-10% with neither Consistency Regularization nor Data Augmentation ( please see our common response to all reviewers at https://openreview.net/forum?id=rJe04p4YDB&noteId=S1lX9YcJsB ). In this setting, Coaching achieves 62.02% top-1 accuracy and outperforms RandomAugment’s 61.88% (see Figure 3, Page 8).
>
> [Experiments with Different Numbers of Labeled Examples]
> Per your concern, we have performed experiments with ImageNet using 20%, 40%, and 80% of labeled data. All models are ResNet-50. The top-1 accuracy for them are as follows:
>
> 20%: 75.41
> 40%: 76.21
> 80%: 76.91
>
> Thus, with 40% of the labeled data, we are around the same ballpark with ResNet-50 trained in a fully-supervised manner using 100% of data. Meanwhile, with 80% of the labeled data in ImageNet, Coaching achieves the same performance with supervised training using all labeled data (whose top-1 accuracy is 76.89%).
>
> [TO BE CONTINUED IN THE COMMENT BELOW]

---

> > ### Author Response · Authors · 2019-11-06
> > **continuing our response above**
> >
> > [On the Comparison of Coaching to Other Baselines]
> > We agree with you that some past semi-supervised learning methods do not use the same data augmentation scheme with ours, and some past methods also do not use consistency regularization. However, we do *not* claim anything about Coaching versus these methods. Quite the opposite, we have mentioned a few times in our paper that we do *not* compare Coaching with these past baselines. For instance, please see our paragraph *Baselines* in Section 3.1 (Page 4) and our Table 1’s caption.
> >
> > In fact, Coaching’s focus is the interaction between the Teacher model and the Student model, and the semi-supervised objectives from previous papers can be applied to our Teacher model to improve the final Student’s performance (as we did with UDA).
> >
> > We understand that controlled experiments are extremely important, and we have followed this procedure. However, it is unreasonable to expect any paper to compare their method to *all* existing semi-supervised learning methods in controlled environments (which requires running *all* methods).
> >
> > We have explained in our paper (Section 3.1, Page 4, paragraph Baselines), the reason why we chose our three baselines. Among them, UDA is already the strongest existing method prior to us, and we showed that Coaching+UDA outperforms UDA by a large margin, *in a controlled setting*.
> >
> > We believe this is sufficient to show the strong benefit of Coaching, but if you have further suggestions on how we can convince you that this is indeed the case, please let us know?
> >
> > [On the Need for the Teacher to Learn from the Student]
> > First, our *controlled* experiments have demonstrated that letting the Teacher to learn from Student’s mistakes leads to stronger performances.
> >
> > Second, we disagree with your point that “Given the strong learning capacity of neural networks, the proposed method will easily be overfitting”. Do we claim that all deep learning methods will easily overfit, just because “the strong learning capacity of neural networks” is presented in *all* neural networks? Of course some methods do, but as we have repeatedly shown in the paper, as well as in our discussion with you above, Coaching is very effective to prevent overfitting.
> >
> > Third, we do *not* fully understand why Coaching prevents overfitting. Our best guess is that since the teacher samples the labels, the student will *not* see the same labels over and over again, which makes it harder to overfit. However, this should be left as a future work, rather than be considered as a reason to reject our paper, especially given all the strong performance that we delivered and *proved* to work.
> >
> > Fourth, using a very simple experiment, we will show that your prediction that “the teacher network will have all labeled data classified correct, and all unlabeled data classified into the same labels as the student network” is wrong.
> >
> > We consider the Four Spins dataset (see https://openreview.net/pdf?id=SyzrLjA5FQ , Section 6.4), which has 10,000 unlabeled examples and 15 labeled examples for each class. We compare Supervised Learning with Coaching (just Coaching, no supervised loss, no consistency loss, etc.), both using a 5-layered perceptron with tanh activations and with hidden layers of 4 units.
> >
> > As can be seen from this screenshot ( https://pasteboard.co/IFlu7uQ.png ), Supervised Learning gets all labeled data point correct and becomes overfit. Meanwhile, Coaching does not overfit, as the resulting student incorrectly classifies some labeled examples. Moreover, Coaching finds a more balanced decision boundary, which should be the case for the Four Spins dataset.
> >
> > Since the Student network only learns from the Teacher's pseudo labels, this result implies that the Teacher does not get all labeled data correct *with high confidence*. Otherwise, the Teacher would only sample the correct pseudo labels for the labeled data and would overfit the student, just like in the supervised learning case.

---

> ### Author Response · Authors · 2019-11-13
> **Please consider increasing your score. We are also happy to discuss more.**
>
> We have responded to many points in your review. Most importantly:
>
> 1. Coaching does not overfit (originally shown in our paper, as well as in further experiments in our response to you).
>
> 2. Coaching’s strong performance is not *only* because of our “Additional Implementation Details” (UDA vs UDA+Coaching in our paper is a controlled experiment. We also provided more controlled experiment in our response to you).
>
> We believe our response has addressed all reservations that you expressed.
>
> Therefore, would you consider increasing your rating for our paper?
>
> Should you have further reservations of our method, we hope we can engage in a discussion.

---

### Official Review · AnonReviewer3 · 2019-10-29
**Official Blind Review #3**

**Rating:** 3

**Review:**

The paper proposes Coaching for semi-supervised learning. The teacher generates pseudo labels for unlabeled data and the performance of the student on labeled data is used as a reward to train the teacher. The empirical results are very impressive.

Overall, the paper is clear and easy to follow. The idea looks interesting to me. However, I find that there are some weaknesses.

First, the method is not well-motivated in the Introduction section.

Second, the derivation of the teacher's update rule is incorrect. In Eq. (11), $g_S^{(t)}(\hat y_{unl})$ is a vector, $\frac{\partial \ell(x_{unl}, \hat y_{unl}; \theta_T)}{\partial \theta_T}$ is also a vector. What do you mean by multiplying two vectors? The left side of Eq. (11) is a matrix while the shape does not match on the right side.  It is incorrect to get Eq. (3) from Eq. (11), especially the transposed $g_S^{(t)}$. Where does the transpose come from?

And $g_T^{(t)}$  in Eq.(3) is inconsistent with line 6 in Algorithm 1, where $\eta h^{(t)}$ is not included in $g_T^{(t)}$.

For experiments, I wonder what is the performance of pure Coaching without RandomAugment and the consistency loss/UDA in Table 1. I think this is a fair comparison with the baselines like Mean Teacher and VAT. I suggest adding this to the ablation study as well.

And I expect more explanations on the "additional implementation details". For example, why is it correct to use cosine distance instead of the dot product? In this case, is it a valid gradient? Same for other tricks. We need to understand why it works apart from adding a bunch of tricks together. And I doubt whether the improvement is due to these tricks or the method itself.

I would be willing to increase the score if all the concerns are addressed in the authors' response.


**Experience Assessment:**

I have published one or two papers in this area.

**Review Assessment: Checking Correctness Of Derivations And Theory:**

I carefully checked the derivations and theory.

**Review Assessment: Checking Correctness Of Experiments:**

I carefully checked the experiments.

**Review Assessment: Thoroughness In Paper Reading:**

I read the paper thoroughly.

---

> ### Author Response · Authors · 2019-11-06
> **To Reviewer 3: Our derivations are correct, and we try to be clearer. We also add mores results and insights.**
>
> Thank you for time and your comments. Below, we try to address your concerns about our paper.
>
> [On the Teacher’s Update Rules]
> We strongly appreciate the fact that you went through the paper, even to the Appendix to figure out the derivation of the Teacher’s update rule.
>
> The update rule is actually correct, but the presentation might be confusing. We apologize for causing you the confusions. We believe such confusions are due to the interchangeable usage of the gradient/Jacobian notations.
>
> To avoid these confusions, we have uploaded a revision of our paper, where we clearly stated which mathematical notations are used. We also annotated the dimensions of the quantities in the equations throughout the derivation. We hope the new version is easier to follow and verify.
>
> [On the Use of Cosine Distance]
> We explain why the use of cosine distance *makes sense*. Now that we have established that Equation 3 is correct, we see that the dot product between two gradients is simply a scalar, which is subsequently multiplied with the teacher’s policy gradient.
>
> The sign of this scalar governs whether the teacher should increase or decrease its confidence in the pseudo labels that it samples. Replacing dot product by cosine distance does not change this sign.
>
> The magnitude of this scalar governs how far the teacher should follow a particular direction. Replacing dot product by cosine distance ensures that the magnitude is <= 1. This makes the updates more stable.
>
> If we wish not to use the cosine distance, we might as well use a learning rate that is a few orders of magnitude smaller. (this is our rough estimation from the fact that many ResNet-50 models in our experiments have the gradient norm of between 300 and 400).
>
> [On the Performance of Coaching without RandomAugment and Consistency Loss/UDA]
> Please see our common response to all reviewers ( https://openreview.net/forum?id=rJe04p4YDB&noteId=S1lX9YcJsB ).

---

> ### Author Response · Authors · 2019-11-13
> **Please consider increasing the score. We're happy to discuss more**
>
> In your review, you wrote: “I would be willing to increase the score if all the concerns are addressed in the authors' response”
>
> In response to your review, we have:
> 1. Updated the paper so that the maths become easier to follow. We have also verified the correctness of the derivation of the teacher’s update rules.
>
> 2. Provided the experiments which demonstrate that the benefit of Coaching is not only because of our “Additional Implementation Details”.
>
> Based on our response, would you consider increasing your rating on our paper? Is there anything else that you want to see to be more convinced of our results?

---

### Author Response · Authors · 2019-11-06
**Common Response to All Reviewers**

We thank the reviewers for their time and comments.

Here, we address the common points to the reviewers.

[On the Performance of Coaching without RandomAugment and Consistency Loss]
Both Reviewer 3 and Reviewer 2 are concerned about whether the strong performance of Coaching comes from the Additional Implementation Details (Section 3.1 in the paper), or from the method by itself.

Below, we present the pure coaching results (no RandomAugment, no Consistency Loss). We do use the cosine distance, the moving average baseline, and the supervised loss on the teacher. All hyper-parameters are the same as in Table 6 in our paper.

  CIFAR-10 (4000)	|	SVHN (1000) 	|	ImageNet (10%)
------------------------------------------------------------------------------
    86.11 ± 0.17 	|	94.21 ± 0.24	|	62.02 / 85.91

Compared to our controlled experiments in Table 1, it can be seen that Coaching outperforms *all* baselines, except for UDA. On ImageNet-10%, Coaching outperforms RandomAugment (60.88 top-1 accuracy; see Figure 3, Page 8). These results imply that the gain delivered by Coaching is non-trivial, and cannot be accounted to *only* our "Additional Implementation Details" (Section 3.1, Page 4).

[Coaching without Supervising the Teacher]
Without the supervised loss, on CIFAR-10, Coaching takes *4 million steps* (wich a batch size of 128) to reach an accuracy of 85.84%. This number is close to the mean performance of 86.11% which is achieved with 1 million steps. This result suggests that without the supervised loss on the teacher, Coaching takes *very long* to converge, but the final accuracy is still in the same ballpark. Since this experiment takes very long, we did not perform it for SVHN and for ImageNet, nor did we repeat it multiple times for mean/std.

Our intuition here is that the Coaching loss in our Teacher model is essentially an on-policy loss in Reinforcement Learning. Similar to many RL systems, such as AlphaGo, AlphaZero, AlphaStar, without some supervised signals, Coaching takes much longer to converge. In fact, for datasets with a large number of classes such as ImageNet, Coaching alone might not converge at all, or takes prohibitively long to converge.

That said, virtually all existing semi-supervised learning methods have a directly cross-entropy loss signal on labeled data, so Coaching is not requiring anything unreasonable.

[Improved Results on ImageNet-10%]
By coaching for *1 million steps* on ImageNet-10% (with a batch size of 2048), we achieve the top-1 accuracy of 73.32% before finetuning on the labels, which improves to 74.01% after finetuning on the labels. This is a new state-of-the-art on ImageNet-10%, outperforming the existing result of 73.21% [1]. Note that [1] uses a 4x wider ResNet-50, while we only use a ResNet-50.

This new result also establishes that Coaching does *not* overfit, as concerned by Reviewer 2. In fact, a ResNet-50 trained on ImageNet-10% (with the same batch size of 2048) would overfit after about 50K steps, as shown in this screenshot ( https://pasteboard.co/IFkOBbz.png ).

Finally, the comparison between Coaching+UDA and UDA is a *controlled* experiment, as UDA has all implementation details with Coaching+UDA. As Coaching+UDA outperforms UDA by a large margin, this result asserts the benefit of Coaching.

[1] S4L: Self-Supervised Semi-Supervised Learning. Xiaohua Zhai, Avital Oliver, Alexander Kolesnikov, Lucas Beyer. https://arxiv.org/pdf/1905.03670.pdf

---

### Decision · Program_Chairs · 2019-12-19

**Decision:**

Reject

**Comment:**

Authors propose a new method of semi-supervised learning and provide empirical results. Reviewers found the presentation of the method confusing and poorly motivated. Despite the rebuttal, reviewers still did not find clarity on how or why the method works as well as it does.